# Mo' Data, Mo' Problems: How data composition compromises data scaling properties in machine learning

## Abstract

The accumulation of data in the machine learning setting is often presented as a panacea to address its many modeling problems—including issues with correctness, robustness, and bias. But when does adding more data help, and when does it hinder progress on desired model outcomes? We model data accumulation from multiple sources and present analysis of two practical strategies that result the addition of more data degrading overall model performance. We then demonstrate empirically on three real-world datasets that adding training data can result in reduced overall accuracy and reduced worst-subgroup performance while introducing further accuracy disparities between subgroups. We use a simple heuristic for determining when the accumulation of more data may worsen the issues the additional data is meant to solve. We conclude with a discussion on considerations for data collection and suggestions for studying data composition in the age of increasingly large models.

## 1 Introduction

The accumulation of data (labeled or unlabeled) in machine learning is often touted as the reliable solution to many of its modeling problems. The benefits of more data on performance have been observed across many domains including tabular (Chen et al., 2018), language (Brown et al., 2020), vision (Chen et al., 2020), and multi-modal data (Wang et al., 2021). Beyond accuracy, increasing dataset size has also been shown to improve adversarial robustness (Carmon et al., 2019) and robustness against distribution shift (Miller et al., 2021). Furthermore, when adding more data also improves subgroup representation, group-level disparities in classification can also be reduced (Rolf et al., 2021). However, practically acquiring more data for training involves much more than a naive increase in the number of training samples. In this work, we define this goal of acquiring more training data examples as *data accumulation*. In practical situations, those engineering the system scale sources to supplement an existing training set – thus data accumulation must not only consider dataset size but also how the composition of the accumulated data changes with scale. Even though such considerations are common knowledge in practical machine learning engineering (Shankar et al., 2022), these challenges still remain relatively under-explored theoretically and empirically by the machine learning research community.

Moreover, dataset qualities do not exist in isolation; algorithmic techniques have been developed for addressing dataset limitations in the distribution shift and supervised domain adaptation literature (Kouw & Loog, 2018). However, current works in these areas narrowly focus on model-based interventions to improve model performance and are thus inadequate for precisely characterizing the effects of a broader set of dataset properties on model outcomes. Acknowledging the reality that there is often agency in the design and composition of the training dataset is an opportunity to design downstream model properties through decision-making about the data. In fact, our paper joins a growing line of work focused on data interventions (Gadre et al., 2023; Marion et al., 2023; Compton et al., 2023). Data accumulation is thus a data-oriented alternative perspective to complement current work which remains narrowly anchored to model improvements.

In this paper, we take a pragmatic approach to data accumulation and construct scenarios that more explicitly factor in corresponding changes to data composition (Figure 1). Motivated by statistics

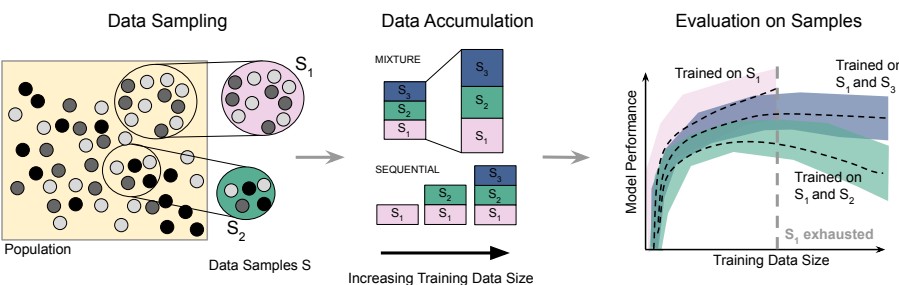

**How does data composition impact model performance under scaling?**

Figure 1: Illustrative pipeline of how we consider the effects of data composition on data scaling properties. We hope to understand cases where the addition of more data in model training leads to a degradation in overall model performance.

literature on the performance penalties incurred by scaling data under sampling bias (Meng, 2018), we explore how this phenomenon may arise in the machine learning setting. We take a principled approach by first formalizing models of data accumulation which gives intuition for why increasing the dataset size may not be sufficient to guarantee better performance. We then test our theoretical models of data composition by examining the effects of increasing training dataset size on real-world tabular datasets for predicting census income, restaurant review sentiment, and medical outcomes.

Our contributions are as follows. In this work, we:

1. **Model realistic case studies of data composition changes in data accumulation:** We present models for data composition changes that occur due to common strategies at unilateral increases to dataset size (ie. scaling up training set size). We motivate and formalize cases of data accumulation from a single-source and multi-source setting.

2. **Analyze data accumulation impacts on downstream performance:** We theoretically demonstrate how data scaling can lead to worse model outcomes and present a simple heuristic to determine when to add more data. We show that under differing sampling regimes, there are scenarios where data accumulation can worsen model performance.

3. **Demonstrate empirical results on the trade-offs between scale and data composition:** We discuss the performance impact of two different practical strategies for data accumulation in the multi-source setting — a sequential data addition case and that of scaling up a mixture of data sources. We illustrate on 3 real-world datasets how the mechanism through which data accumulation occurs impacts model properties.

Most importantly, we hope for this work to be a critical starting point in formalizing the complex dynamics underlying data decision-making as part of the machine learning process. The details of data practices are often overlooked by the machine learning research community altogether, despite its key role in determining the nature of model outcomes (Paullada et al., 2021). We hope this work can be a strong starting point for a deeper investigation by the machine learning community into more principle-based foundations of meaningful data practices.

## 2 RELATED WORK

**Data scaling laws influence model outcomes** "Scaling laws" more broadly refer to how increases in model "size" lead to improved performance. Typically, model size is described in terms of the number of model parameters, compute, and other measured factors characterizing the model (Kaplan et al., 2020; Bahri et al., 2021). Data "scaling laws" (Zhang et al., 2020; Bansal et al., 2022; Zhai et al.) specifically reveal the way in which training on larger and larger datasets yields improved performance. Furthermore, adding more data has been suggested as a way to improve the fairness of a model (Chen et al., 2018).

Figure 2: Our work on data accumulation characterizes training data properties interact with scale and how they impacted model outcomes. This study is data-centric and is orthogonal but complimentary to model-based interventions provided in adjacent directions such as domain adaptation.

**Data composition influences model outcomes** However, the composition of training data, at any size, has been shown to influence model outcomes. Data properties such as data diversity, redundancy, and noise can all contribute to model performance, robustness, fairness, and efficiency Mitchell et al. (2022). These data properties are typically determined by how the data is collected. For example, Rolf et al. (2021) suggests sampling directly from group-specific distributions in order to improve model performance on certain under-represented subgroups.

Prior work in domain adaptation and distribution shift has also considered data composition independent of size (Kouw & Loog, 2018; Gulrajani & Lopez-Paz, 2020; Yang et al., 2023). However, in these works, training data is often considered to be set in stone and imposed as a pre-existing and static constraint. Thus, many of the challenges incurred by data composition are typically addressed with algorithmic interventions rather than data-centric decision-making.

**Realistic data scaling impacts data composition** When realistically increasing the size of a training dataset, compositional changes in the data can be introduced. Thus, a more pragmatic perspective to data scaling that factors in changes to overall data composition is required – we call this process *data accumulation* (Figure 2). In surveys, sampling bias exacerbates mis-estimation error as the sample size increases, as observed in settings estimating vaccine uptake (Bradley et al., 2021) and election polling (Meng, 2018).

Thus far, relatively few works in machine learning have critically examined the effect of increasing training data size while factoring in the potential changes to data induced by scaling. In the image classification setting, recent work found that performance heavily depends on the pre-training source data (Nguyen et al., 2022) and spurious correlations may be introduced when combining data sources (Compton et al., 2023). Using a theoretical model, Hashimoto (2021) looks at data as a fixed mixture of different sources (e.g., different categories of Amazon reviews) to characterize excess loss as dataset size increases.

## 3  TWO MODELS FOR DATA SCALING

Much of the past work on the impact of data scaling on model performance assumes a *single-source setting*, where data from a single data sampling process is scaled up, and the distribution of the dataset remains fixed. It is under this setting that many claims on data scaling are typically considered. However, in most practical settings, data accumulation occurs in a *multi-source setting*, where the final training dataset is pieced together from multiple distinct data sampling processes. In order to practically increase the size of the training set, data from multiple sources are collected and combined. Unlike the single source setting, there is not one static scaled-up data collection process – instead the resultant larger dataset is a mixture of multiple data sources, and thus presents more complex data composition changes as the dataset size increases.

In this paper, we specifically consider two practical approaches of this multi-source setting: the SEQUENTIAL case and the MIXTURE case. We consider the following scenarios: 1) enlarging the dataset by sampling from a mixture of fixed sources and 2) sequentially adding data across different sources for model training. The key observation we make in this paper is how overall and subgroup performance can vary as we increase the sample size $n$ in both of these scenarios.

**Preliminaries** Let $\{x, y\}^n \sim \mathcal{D}$ be data generated from some underlying distribution. Let $D_{S_1}, ..., D_{S_m}$ be a series of empirical distributions sampled with varying types of sampling bias from the underlying distribution $\mathcal{D}$; these empirical source distributions are finite and are different fixed sizes $n_{s_1}, ..., n_{s_m}$. The training distribution of sized $n$, $D_{train,n}$, is composed of these $m$ sources. The deployment distribution $D_{test}$, where we hope to achieve good performance, is sampled directly from $\mathcal{D}$ without any sampling bias. We define $\delta : \mathcal{P}(\mathcal{X}, \mathcal{Y}) \times \mathcal{P}(\mathcal{X}, \mathcal{Y}) \to \mathbb{R}_{\geq 0}$ as a divergence between distributions.

**Mixture Case (MIXTURE)** In the mixture case, $D_{train,n}$ comes from a mixture of sources: $D_{train,n} = \sum_{i=1}^{m} \alpha_i D_{S_i}$. Here, the coefficients $\alpha_i \in [0, 1]$ ($\sum_i \alpha_i = 1$) specify what proportion of data points are sampled from each source. Given a fixed vector $\alpha$ and a dataset size $n$, $n \times \alpha_i$ data points are included from each distribution $D_{s_i}$; the ratio of sources is independent of $n$. Since the ratio of sources is fixed, we also expect the divergence between train and test distributions $\delta(D_{train,n}, D_{test})$ to remain constant as $n$ increases.

**Sequential Case (SEQUENTIAL)** In the sequential case, the training data is a strictly increasing collection of the underlying sources: $\hat{D}_{train,n} = (\bigcup_{i=1}^{k-1} \hat{D}_{S_i}) \cup \frac{n - \sum_{i=1}^{k-1} n_{s_i}}{n_{S_k}} \hat{D}_{S_k}$[1]. Here, $k$ is set to the source index such that $\sum_{i=1}^{k-1} n_{s_i} < n \leq \sum_{i=1}^{k} n_{s_i}$. In other words, for a desired dataset size $n$, we start by adding data from the first source and continue to add data from sources sequentially until we reach $n$. This addresses the common scenario where acquiring more data incurs additional cost; all data from existing sources are used before a new source is introduced. The resulting distribution can also be viewed as a mixture of source distributions where $\alpha$ depends on $n$: $D_{train,n} = \sum_{i=1}^{k} \alpha_i D_{S_i}$ where $\alpha_i = \frac{n_{s_i}}{n}$ for $i < k$ and $\alpha_m = \frac{n - \sum_{i=1}^{k-1} n_{s_i}}{n_{s_m}}$.

A key observation we make is that $\delta(D_{train,n}, D_{test})$ in the sequential case will actually depend on the number of samples.

**Example 3.1.** *Consider a training set of two sources: $D_{S_1}$ a small high-quality dataset, $D_{S_2}$ a large lower-quality dataset. We can model the divergence between train and test distributions as follows **if** $\delta$ composed linearly:*

$$\delta(D_{train,n}, D_{test}) = \begin{cases} \delta(D_{S_1}, D_{test}) \text{ if } n \leq |D_{S_1}| \\ \frac{|D_{S_1}|}{n} \delta(D_{S_1}, D_{test}) + (1 - \frac{|D_{S_1}|}{n}) \delta(D_{S_2}, D_{test}) \text{ otherwise} \end{cases}$$

While we cannot assume that divergences compose linearly, we can limit our scope to $f-$divergences and use the convexity of this class of divergences to show that in the SEQUENTIAL case, $\delta(D_{train,n}, D_{test})$ might increase with $n$.

**Lemma 3.2.** *Let $D_{train,n}$ be constructed in the SEQUENTIAL case from $k$ sources: $D_{S_1}, ..., D_{S_k}$, then if $\delta(D_{S_k}, D_{test}) - \frac{cn}{n_{s_k}} \geq \delta(D_{train,n}, D_{test})$ :*

$$\delta(D_{train,n}, D_{test}) \geq \delta(D_{train,n-n_{s_k}}, D_{test})$$

*where $\delta$ belongs to the family of f-divergences and $c$ is a divergence-dependent constant where $\delta(D_{train,n}, D_{test}) + c = \sum_{i=1}^{m} \frac{n_{s_i}}{n} \delta(D_{S_i}, D_{test})$.*

Lemma 3.2 gives a relationship between the new data source and the test set that would cause the divergence between train and test distributions can increase with $n$ in the SEQUENTIAL case[2]. In a fixed dataset size setting, Acuna et al. (2021) relates increased divergence to empirical risk for $f-$divergences in particular by giving a generalization bound. Prior works have also considered different discrepancy measures including $L_1$ distance (Ben-David et al., 2006), $\mathcal{H}\Delta\mathcal{H}$ divergence (Ben-David et al., 2010), margin disparity discrepancy (Zhang et al., 2019). In conjunction with Lemma 3.2, these results from prior works give an intuition for a larger empirical risk upper bound when the divergence between train and test distributions increases. However, what remains unanswered is whether this bound remains while the training set size increases.

---

[1] $\hat{D}$ denotes the *set* of examples or data points in the distribution D

[2] See proof in the appendix

| DATASET | NUMBER OF ROWS | OUTCOME | SOURCE | SUBGROUP |
|---|---|---|---|---|
| Folktables (Ding et al., 2021) | 1,664,500 | Binary Income Level | State | Race |
| Yelp (Yelp, 2023) | 6,990,280 | Multi-Class Review Stars | State | Restaurant Category |
| MIMIC-IV (Johnson et al., 2020) | 197,756 | Binary Readmission | Admission Type | Race |

Table 1: Dataset overview for experiments.

Using the two cases of data accumulation, we can reason about what we might observe empirically even if we do not have an oracle divergence metric. As the training dataset size increases in the MIX-TURE case, we would expect $\delta(D_{train,n}, D_{test})$ to remain constant and the training loss to decrease. Thus, we expect the upper bound of the test loss to become tighter as the dataset size grows. However, in the SEQUENTIAL case, if there is a combination of sources that causes $\delta(D_{train,n}, D_{test})$ to grow faster than the decrease in the training set risk, this upper bound becomes looser and we may see an increased test set risk.

### 3.1 CRITERIA FOR REJECTING MORE DATA

We consider the SEQUENTIAL case where a practitioner may have trained on some data $D_{train}$ to obtain some predictor $f_{D_{train}}$, and encounters another data source $D_i$. The question is whether it would be beneficial to enlarge the dataset and train on $D_{next} = D_{train} \cup D_i$ in order to best perform on the test distribution that we care about $D_{test}$. More formally, we are concerned with the excess risk under some proper loss function $l$ from adding more data:

$$L(f_{D_{train}}, f_{D_{next}}, D_{test}) = \mathbb{E}_{(x,y) \sim D_{test}} \left[ l(f_{D_{train}}(x), y) - l(f_{D_{next}}(x), y) \right]$$

If $L(f_{D_{train}}, f_{D_{next}}, D_{test}) > 0$, we would want to incorporate the additional data to achieve a lower risk on the test distribution. Otherwise, we would reject the additional data in order to not increase risk on the test distribution or to avoid incurring extra costs for data access and model training while not improving the risk. Practically, finding $f_{D_{next}}$ already requires training a model based on an additional dataset. To avoid this, we suggest a rejection criterion based on access to the existing data mode $D_{train}$, the additional data $D_{next}$, and the test distribution $D_{test}$. While in reality the test set cannot be accessed, we assume we can use some part of the training set (e.g., the $D_{s_1}$) that is similar to the test distribution). Prior works have suggested that measuring excess Kullbeck-Leibler (KL) divergence between train and test distributions correlates with the resulting loss (Xie et al., 2023). We also use the excess KL as a heuristic:

$$\Delta_{KL}(D_{train}, D_{next}, D_{test}) = \delta_{KL}(D_{next}||D_{test}) - \delta_{KL}(D_{train}||D_{test}) \tag{1}$$

## 4 EXPERIMENT SETUP

**Datasets** For our investigation, we study three real-world tasks and datasets, chosen because of their rich feature sets and the open-access availability. Dataset details can be found in Table 1. See appendix for code, additional experiments and details, and synthetic data experiments.

**Models and Evaluation** We consider the scenario where the initial dataset of interest comes from a single data source, i.e., Source A, with limited training examples (e.g., South Dakota (SD) in Folktables). We also consider at least one second available data source, i.e., Source B, from which additional training examples can be drawn. The experimental goal is to investigate the effects of manipulating training data composition on model outcomes as measured on a test set sampled exclusively from the initial dataset Source A (e.g., SD) – which we call the *reference test set*. Further experiments are also evaluated on a *generalized test set*, which is randomly sampled from a mixture of all available data sources.

To observe data accumulation in the MIXTURE case, Source A and Source B are sampled at a fixed ratio from a combined dataset of Source A and Source B in order to increase the training set size.

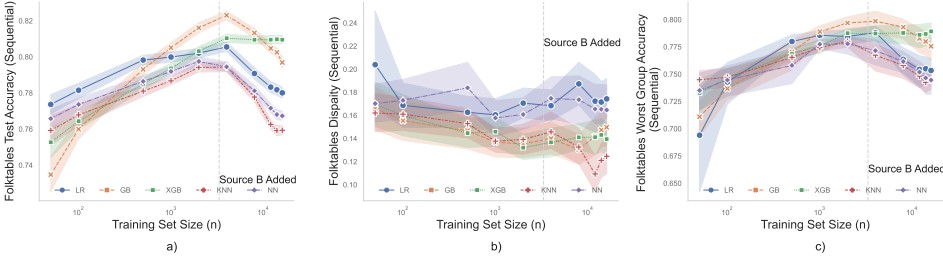

Figure 3: Folktables results on **Source A reference test set** in the SEQUENTIAL case over 5 trials on **(a)** accuracy, **(b)** accuracy disparity, and **(c)** worst subgroup accuracy. Source A from South Dakota and Source B from California is added once South Dakota data has been exhausted.

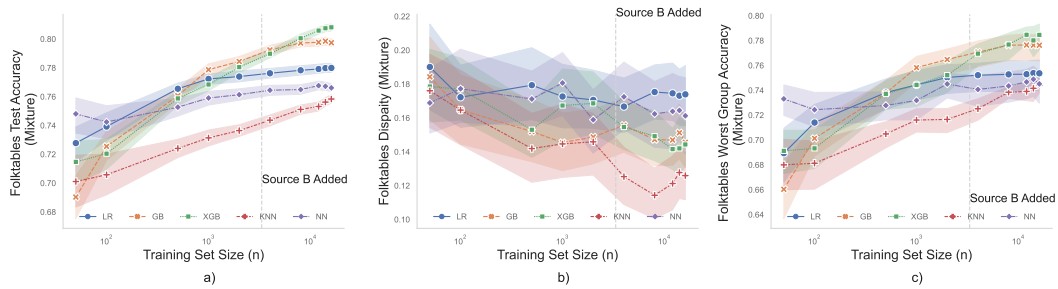

Figure 4: Folktables results on **Source A reference test set** in the MIXTURE case over 5 trials for **(a)** accuracy, **(b)** accuracy disparity, and **(c)** worst subgroup accuracy. The mixture is the same ratio as the final dataset for the SEQUENTIAL case in Figure 3) (75% CA and 25% SD)

In the SEQUENTIAL case, we start by adding training data from Source A and then from Source B when all points from Source A have been included.

We consider a variety of different models: logistic regression (LR), gradient boosting (GB) (Friedman, 2001), k-Nearest Neighbors (kNN), XGBoost (XGB) (Chen & Guestrin, 2016), and MLP Neural Networks (NN). Let $f$ denote the model we are evaluating and let $g$ be a group function that maps each data point to a subgroup, we evaluate the following metrics over $D_{test}$: **Accuracy**: ($\mathbb{E}_{(x,y)\sim D_{test}}[f(x) = y]$), **Disparity**: The difference between the best and worst-performing subgroups ($\max_{g'} \mathbb{E}_{(x,y)\sim D_{test}}[f(x) = y|g(x) = g'] - \min_{g'} \mathbb{E}_{(x,y)\sim D_{test}}[f(x) = y|g(x) = g']$), and **Worst group accuracy**: The accuracy on the worst-performing subgroup and the metric of interest in studying subpopulation shifts in the distribution shift literature (Koh et al., 2021) ($\min_{g'} \mathbb{E}_{(x,y)\sim D_{test}}[f(x) = y|g(x) = g']$).

## 5 RESULTS AND ANALYSIS

**Single Source Datasets Benefit from Data Scaling Properties**   We consider the initial stage of the SEQUENTIAL case—prior to sampling from any additional data sources—to be equivalent to a single-source data setting. We find that increasing the dataset size in this single source setting yields improved performance (Figure 3a). Consistently, maximum accuracy is achieved when the most data points in a single source are used. Single source data increases also improve worst-subgroup performance, as demonstrated in prior literature (Sagawa et al., 2019). For some models, increases in data slightly improve disparity (e.g., XGB disparity drops from 17.0% to 13.6%) while for other models even within the single source setting do little to minimize the differences between subgroup disparity[3].

---

[3]Results with confidence intervals and additional results in larger $n$ for single source scaling are available in the appendix

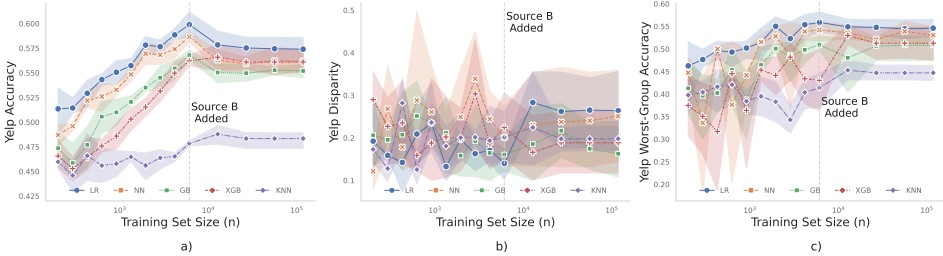

Figure 5: Yelp results on **Source A reference test set** in the SEQUENTIAL case over 5 trials on **(a)** accuracy, **(b)** accuracy disparity, and **(c)** worst subgroup accuracy. Source A is from New Jersey and Source B is from Pennsylvania.

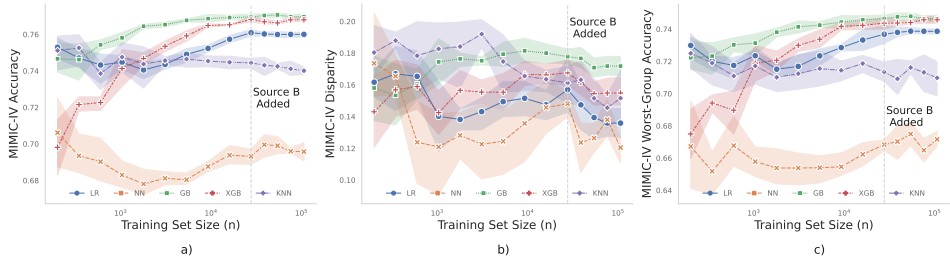

Figure 6: MIMIC-IV results on **Source A reference test set** in the SEQUENTIAL case over 5 trials on **(a)** accuracy, **(b)** accuracy disparity, and **(c)** worst subgroup accuracy. Source A is from admission type URGENT and Source B is from EW. EMER.

**Multi-source Dataset Scaling Can Lead to Worse Outcomes on a Reference Test Set**    In the SEQUENTIAL case (Figure 3a, 5a, 6a), we observe that adding additional data from a separate source, thereby quadrupling the size of the training set, leads to a dip in performance on a reference test set of interest, in addition to worse fairness metrics and worse robustness. For the Folktables dataset (Figure 3a), this dip is observed empirically as a statistically significant reduction of test accuracy for all models except XGB (e.g., LR: -2.5%; GB: -2.6%; kNN: -3.5%, NN: -2.7%). This reduction in performance occurs when additional data is added from source B ($n_B = 12000$) once source A ($n_A = 4000$) is exhausted; the training data size has tripled. Decreases to worst subgroup performance were also significant and observed in all models except XGB (e.g., LR: -3.5%; GB: -2.3%; kNN: -2.3%, NN: -2.7%). We did not observe a significant decrease in disparity with the addition of more data.

In the MIXTURE case (Figure 4), we find that scaling a fixed mixture of sources yields monotonically improved performance on the reference test set—i.e., there is no observed dip in performance, and the increase of the dataset size is correlated with increasing test accuracy, and better worst subgroup performance. From $n = 4000$ to $16000$, we see test accuracy consistently improve for all models (e.g., XGB:+1.8%; kNN:+1.5%) and we see worst subgroup performance also consistently improve across all models. As in the SEQUENTIAL case, we do not see a significant impact of data scaling on subgroup disparities due to the high variance in worst and best subgroup size in the test set. However, comparing the MIXTURE case to the SEQUENTIAL over the same range of $n$ before all data is added, the accuracy achieved on the reference dataset for the best $n$ in the SEQUENTIAL case remains consistently above the maximum accuracy in the MIXTURE case. At $n_{max}$ for SEQUENTIAL case, test accuracy is higher (LR: 78.0%, GB: 79.8%, XGB: 80.8%, kNN: 75.8%, NN:76.7%) than $n_{max}$ for the MIXTURE case (LR: 80.6%, GB:82.3%, XGB:81.0%, kNN:79.4%, NN: 79.6%).

**Multi-source Dataset Scaling Can Lead to Better Generalization**    In (Figure 5b, 6b, 7a), we see that increases to the training dataset, even from sources of different distribution, still yield improvements on a generalized test set, sampled from all available sources (e.g., For $N = 50/4000/16000$, LR: 72.7% / 76.8% / 78.4%; GB: 70.4% / 78.9% / 80.2%; XGB: 72.1% / 79.5% / 80.5%; NN: 70.8% / 77.4% / 77.9%). This indicates that increasing dataset size across sources yields improvements to

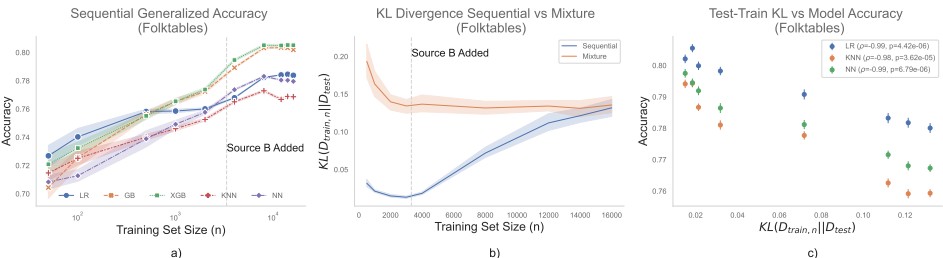

Figure 7: **(a)** Folktables accuracy in the SEQUENTIAL case changes on the **generalized test set**, **(b)** Comparison of Train-Test KL: SEQUENTIAL setting vs MIXTURE setting, **(c)** Relationship between Train-Test KL and Accuracy in the SEQUENTIAL setting for different classifiers

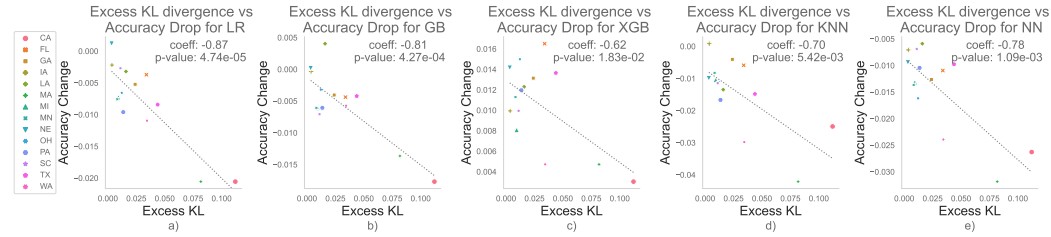

Figure 8: The relationship between Excess KL and the resulting accuracy drop from increasing for **a)** logistic regression (LR), **b)** gradient boosting (GB), **c)** XGBoost (XGB), **d)** K-Nearest Neighbor (kNN) and **e)** Neural Network (NN). We observe a statistically significant correlation between Excess KL (Equation 1) and accuracy drop across all 5 models where we observe significant decreases in model performance.

model generalization, even if this may not translate to improved performance on the reference test set of interest.

## 5.1 PERFORMANCE VIA THE LENS OF A PRACTICAL DIVERGENCE

In Section 3, we presented two models: SEQUENTIAL and MIXTURE. We will now connect our empirical results to our proposed theoretical models of data accumulations.

**Divergence comparison: SEQUENTIAL VS MIXTURE** The first step is to empirically validate that our specific choice of divergence, KL divergence, increases in the SEQUENTIAL setting as the training set size grows. We approximate densities through kernel density estimation with a Gaussian kernel on scaled PCA projections (3 components). Figure 7b compares the KL divergence between the training set and test set at different values of $n$ (data size) for the Folktables Income dataset. In the SEQUENTIAL case, scaling up $n$ results in an increase in train-test divergence while in the MIXTURE case, this divergence remains static.

**Translating Divergence to Accuracy** The next step is to validate that increased train-test divergence translates into a reduction in accuracy. We find a significant negative correlation between the KL divergence between the train and test dataset with the resulting model accuracy for 3 out of 5 models; as train-test divergence increases, test accuracy decreases. Figure 7c shows this correlation for the 3 algorithms where we observe a significant correlation. There was also a negative correlation between for Gradient Boosted Trees (GB). We did not observe a decrease in performance for XGBoost (XGB), thus such a correlation for the XGB model is not expected.

**Excess KL and Rejecting More Data** Finally, we validate our proposed heuristic of excess KL ($\Delta_{KL}$) for deciding when to include more data (Section 3.1). If $\Delta_{KL}$ is larger than $0$ there is a significant distribution shift in the larger dataset and there is thus likely to be an

increase or flat-lining of loss. We consider a large set of states as additional data sources: some are closer to South Dakota (e.g., Minnesota) while others are very distant (e.g., Florida).

When comparing excess KL between the new bigger training distribution and the original dataset relative to the reference dataset, we find a significant negative correlation between accuracy change and excess KL across different states for all the classifiers. These results show that excess KL is indeed a reasonable heuristic for estimating the accuracy drop induced by additional data. Furthermore, the relative ordering of source states in terms of accuracy drop remains consistent across classifiers. However, the scale of accuracy drop for XGB is an order of magnitude smaller than other classifiers. Our results suggest that more data, albeit from a different distribution, does not affect XGB adversely to the same degree.

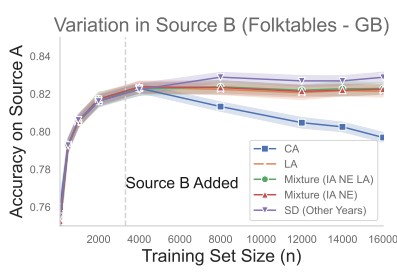

Figure 9: Accuracy on source A when different additional sources are used.

If we replace California with a different state to be the additional data source based on excess KL, we observe better performance as more data (Figure 9). Surprisingly, using Louisiana alone is as helpful as using a mixture of states near South Dakota (e.g., Nebraska and Iowa). Ultimately, the best improvement in performance comes from using South Dakota data from future years (2015-2018) but this source is only slightly better than using a state across the country (e.g. Louisiana) from the same year [4]

## 6 DISCUSSION

We present two practical cases of data accumulation from multiple sources. We observe a decline or flat-lining in overall accuracy and worst subgroup accuracy when we add data in the SEQUENTIAL case in 3 real world datasets. Since the SEQUENTIAL case can be widespread, we urge caution when enlarging training datasets in real-world data collection. We use excess KL divergence as a heuristic for estimating when adding more data might be undesirable. Ultimately, we expect the trade-off between a smaller high-quality data set and a larger low-quality dataset to be model-dependent. Nevertheless, our results motivate practitioners, particularly in high-stakes applications, to carefully consider the costs and benefits of adding more data.

**Limitations & Future Work** For future work, there are many opportunities to investigate data-driven performance trade-offs under more complex data accumulation and data curation schemes, including cases involving data pruning (Sorscher et al., 2022; Hooker et al., 2020) and synthetic data (Nikolenko, 2019) or settings that are a combination of the MIXTURE and SEQUENTIAL settings we present. Our work focuses on the tablular data setting – many of the claims made around data scaling laws relate to the context of large language models (Kaplan et al., 2020; Bansal et al., 2022), computer vision models (Zhai et al.) and image-text models (Nguyen et al., 2022) in an over-parameterized regime (Nakkiran et al., 2021), involving much more complex and expressive algorithms (ie. transformers, convolution neural nets). The data scaling claims in the tabular and small-scale setting we investigate may be sufficient illustrations of possible data dynamics but fall short of painting a complete picture of a general law for data accumulation, if one were to exist.

**Conclusion** Data decision-making is a critical factor in the effective execution of machine learning – yet little is understood of how practical data curation and collection strategies ultimately impact model outcomes. This work is an initial inquiry into what is often an overlooked aspect of the machine-learning process. We challenge long-held assumptions around data scaling, revealing the complexity in enlarging dataset size in the practical setting and discussing how this complexity could yield scenarios in which this assumption, that more data is all you need, no longer holds. We hope this can be a vehicle towards more thorough future modeling and investigations into the practical data decision-making process that underscores much of the model behavior in deployed systems.

---

[4]Our metric can also be applied to mixtures of sources added sequentially. We add a discussion of this in the appendix.

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
