# OpenReview forum: "Mo' Data Mo' Problems: How Data Composition Compromises Scaling Properties"
_ICLR.cc/2024/Conference — Submitted to ICLR 2024_

### Official Review · Reviewer_7keK · 2023-10-27

**Soundness:** 3 good
**Presentation:** 2 fair
**Contribution:** 1 poor
**Rating:** 3
**Confidence:** 4

**Summary:**

The work is devoted to exploring the difference between two ways of sampling data from different sources facing shifts in them compared to the general distribution. The authors demonstrate that sequential sampling expectedly results in the shift of performance. The work contains large experimentations resulted in observational study of the effects.

**Strengths:**

- Intensive experimental evaluation and analysis

**Weaknesses:**

I see a notable weakness of this work in the novelty and the contribution. The paper seems to state claims that are more-or-less known in research community. So, even if there are no publications on the topic, e.g., the fact of having shift of distributions (and thus, lower performance) in the case you sample not from the target distribution is common knowledge and directly follows from ML grounds. So, such knowledges are also referred to as folklore. So, in my opinion, the paper contribution is narrowing down to experimental analysis, which is good but looks like an observational study without clear new insights (besides the ones expected by folklore knowledge). It is not enough for this venue. I would assume that such an analysis is a good illustration / help for students that study ML / statistics and might be published through some books on the topic.


Another related weakness is the problem setup. The authors try to explore some effects when sampling not from the underlying distribution D in different ways knowing that they sample with shifts. What is the problem to be solved? If the practitioner face a situation, knowing D, then they will sample from it. If they have no access to D, then they try to sample some distribution such that it is most close to D. If they face two sources and expect that they have representation of D (they believe that their union is close to D), then they will sample randomly from the union. And etc. In any way, the practitioner knowing basics of ML, will attempt to be close to the best knowledge of D for them. It is hard for me to imagine situations described in the work, where a practitioner is aware of ML grounds and is increasing samples without carrying about the general population. I assume, it might be the case when this practitioner is working with ML tools without knowing ML grounds (so, in this case, a book on ML grounds might help). Overall, the work lack of clear problem statement: what is known for a practitioner, what is not, which decisions they can take, what are limitations.


Sec.3.1 serves for me as support of absence of clear problem statement in the work. The authors write “While in reality the test set cannot be accessed, we assume we can use some part of the training set (e.g., the D_s_1 ) that is similar to the test distribution).” Seems that it means that the practitioner believes that D_s_1 IS the distribution D. So, when receiving D_s_2 they should take D_s_1 united with D_s_2 as D, or not considering D_s_2 at all.

**Questions:**

In section 3.1: “While in reality the test set cannot be accessed, we assume we can use some part of the training set (e.g., the D_s_1 ) that is similar to the test distribution).”

-	I see two closing parentheses while having only one open one.

-	What do you mean saying “that is similar to the test distribution” it is unclear


In Section 4: “data composition on model outcomes as measured on a test set sampled exclusively from the initial dataset Source A (e.g., SD) – which we call the reference test set.”

-	I do not understand: at the beginning of the work it was stated that “test set” is the union. Here, it is stated that it is just D_d_1.

---

> ### Author Response · Authors · 2023-11-16
> **Clarification of Our Work and Contributions**
>
> We thank the reviewer for raising their concerns; however, our contributions may have been fundamentally misunderstood by the reviewer.
>
> **Main Contribution**
> > So, even if there are no publications on the topic, e.g., the fact of having shift of distributions (and thus, lower performance) in the case you sample not from the target distribution is common knowledge and directly follows from ML grounds.
>
> The reviewer is correct that the fact that “distribution shift between train and test datasets results in reduced performance” has been studied extensively in past work (See our references Sagawa et. al. in the main text and Appendix C.5 detailed related work on distribution shift).
>
> However, what is **not** known is (1) why distribution shift may arise as dataset size increases and (2) do the harms induced by distribution shift from scaling eclipse the benefits of scaling. This is an incredibly common scenario in the real world. For example, government resource allocation for social programs depends on projections from census data [1]. In smaller counties and states, the decision has to be made on whether adding more data is beneficial or harmful to achieve more accurate and equitable predictions. Moreover, formalizing **why** adding data does not yield better outcomes is important and it is a central contribution to our work.
>
> > What is the problem to be solved? If the practitioner face a situation, knowing D, then they will sample from it. If they have no access to D, then they try to sample some distribution such that it is most close to D.
>
> Without access to D, how is it possible to know what distribution is closest to D? Our central question (Figure 1) is: “**How** does data composition impact model performance under scaling?”. We examine the situation where there is limited access to D (e.g. a very small sample). A problem arises when we have to decide whether adding data from some other distribution $D_{s_2}$, is beneficial or harmful with respect to overall accuracy, worst group accuracy, and group accuracy disparity.
>
> > Seems that it means that the practitioner believes that D_s_1 IS the distribution D. So, when receiving D_s_2 they should take D_s_1 united with D_s_2 as D, or not considering D_s_2 at all.
>
> Suppose we are policymakers in South Dakota, $D_{s_1}$ is the dataset from our state, but $D_{s_2}$ is a very different state like California. If $D_{s_2}$ is not considered at all, $D_{s_1}$ might be too small and uniform to give adequate accuracy/fairness (e,g, only 2.6% of the population is Black [2] ). If we sample from the union of $D_{s_2}$ and $D_{s_1}$, the result would depend on the relationship between $D_{s_2}$ and $D_{s_1}$ (this is captured in our Mixture model Sec. 3).
>
> Crucially, we know that California is a different state but we do not know **how** it is different. In fact, the Mixture model (see Section 3), would sample from the union of South Dakota and California and we see that it yields worse results than just $D_{s_1}$ in the sequential model with a smaller n (Figure 3a vs Figure 4a). We also know that Louisiana could be as different from South Dakota as California is. However, adding data from Louisiana improves outcomes (Figure 9 vs Figure 3a). We identify Louisiana as a useful state to add using the heuristic we propose (Equation 1).
>
>
> **Answers to the reviewer’s questions**:
> > I see two closing parentheses while having only one open one.
>
> This is a typo where there is an extra closed parenthesis symbol that we will update the paper to exclude the second “)”.
>
> > What do you mean saying “that is similar to the test distribution” it is unclear
>
> In our problem setup, the first source $D_{s_1}$ is very small and limited. For example, if you are a very small state (e.g. South Dakota) or a local hospital with a limited number of ICU records. We assume that to evaluate the trade-offs between training on more data vs induced distribution difference, the deployment dataset may be different from the first source but is likely to be most similar to the first source (e.g. $D_{s_1}$).
>
> > I do not understand: at the beginning of the work it was stated that “test set” is the union. Here, it is stated that it is just D_d_1.
>
> In section 3, we state that in the data accumulation setting, the **training set** is a union of sources. Could the reviewer tell us specifically where we said the test set is the union?
>
> We hope our explanations have clarified the novelty of our contributions. As far as we are aware, we agree with the reviewer that there is no prior work that models the mechanisms that induce distribution shift when dataset size is scaled up. Moreover, our theoretical contributions are validated by our empirical results in different settings, and our practical metric bridges our theoretical model with empirically observed divergences.
> We are happy to continue answering any questions to help the reviewer better understand the contributions of our paper.

---

> > ### Author Response · Authors · 2023-11-16
> > **References**
> >
> > **References**
> > [1] Hotchkiss, Marisa, and Jessica Phelan. "Uses of Census Bureau data in federal funds distribution." US Dept. of Commerce, Econ. and Statistics Administration (2017).
> > [2] United States Census Bureau Quick Facts https://www.census.gov/quickfacts/fact/table/SD/PST045222

---

### Official Review · Reviewer_4rSz · 2023-11-01

**Soundness:** 2 fair
**Presentation:** 3 good
**Contribution:** 1 poor
**Rating:** 3
**Confidence:** 4

**Summary:**

The paper introduces models for data composition changes in single-source and multi-source settings, analyzes the effects of data scaling on performance, and presents empirical findings for three real-world datasets. The authors also use a heuristic for determining when the addition of more data could be detrimental based on measuring excess KL divergence. The paper only focuses on tabular data scenarios.

**Strengths:**

- I appreciate the theoretical result provided by the authors as Lemma 3.2.
- The authors evaluate on 3 real-world datasets and conduct multiple diverse experiments throughout the paper.

**Weaknesses:**

In my opinion, there are multiple issues with the work originating from the simplicity of the findings, mismatch in motivation/setting, and a lack of consistent evaluation throughout. I provide more details on these below, but due to these reasons I am leaning towards rejection as I believe the paper does not meet the bar for acceptance at ICLR:

- **Simple Empirical Findings and Lack of Generalizability**: The experiments conducted as well as the results obtained are quite simple. Both the Sequential and Mixture model settings are quite trivial, and the obtained results are unsurprising to me. For instance, it seems intuitive that multi-source dataset scaling could lead to worse outcomes on a reference test set, and that it might lead to better generalization (Section 5). The only results that look at data scaling more deeply are located within Section 5.1, but still by themselves those cannot motivate this work. Furthermore, the results obtained (as well as the approach undertaken) are highly dataset dependent (for e.g., what if I sample the reference test set differently for Folktables? Or use a different state altogether?) This issue is also showcased via the accuracy results for both the Yelp and MIMIC-IV datasets under the Sequential paradigm (refer to Figure 6a and Figure 7a, respectively). These figures (for obvious reasons) show very different trends across both datasets for the same model.
- **Inconsistent Evaluation Across Datasets**: The experiments are mostly conducted on the Folktables dataset, with a few results for Yelp and MIMIC-IV. For consistency in evaluation, all the datasets should be used and conclusions can be drawn from the results more adequately. For instance, all the results for Section 5.1 (such as those on the generalized test set) consider only the Folktables dataset, and Yelp and MIMIC-IV are not considered. Furthermore, it is not always mentioned in the text when the other datasets are being used and what the motivation is to discard others, which can be confusing for readers.
- **Limited Scope and Applicability**: The biggest drawback of the work is the mismatch in whether this work tackles a useful practical problem and its actual motivation. The original outlined motivations in the abstract and introduction imply that the paper will aim to provide more insights on data scaling in useful practical scenarios. However, the work only considers tabular data and very simple models (the most complex is the MLP). The focus on tabular datasets significantly narrows the paper's scope, especially considering that data scaling is a critical concern in large language models (LLMs) and other domains outside of tabular data (such as NLP and Vision). The paper fails to provide insights or implications for these broader and arguably more impactful areas (such as deep learning), limiting its relevance, scope and applicability. In its current form, I do not think the paper provides insights that are useful for a real-world practical application scenario.

**Questions:**

- Could the authors provide an appropriate justification or real-world scenario as an example for concentrating exclusively on tabular datasets? In this scenario, if models are not deep learning based would they be prone to large data scaling issues?
- In this simpler paradigm with models such as Logistic Regression etc, instead of continuously adding more data for generalizability, would it not make sense to just focus on approaches that curb distribution shift and retrain the model cyclically over certain time periods?
- Please feel free to respond to any of the other weaknesses listed above.

---

> ### Author Response · Authors · 2023-11-16
> **Clarification of our contribution and justification of our scope (1/2)**
>
> We thank the reviewer for reading through our paper and raising interesting questions. We would like to reiterate the contribution and scope of our paper in our response and clarify some misconceptions.
>
> **Answers to the reviewers’ questions**:
> > Could the authors provide an appropriate justification or real-world scenario as an example for concentrating exclusively on tabular datasets? In this scenario, if models are not deep learning based would they be prone to large data scaling issues?
>
> Tabular datasets remain fundamental in many applications such as surveys (e.g. census, public health), loan decisions, recidivism prediction, portfolio optimization, and recommender systems. For fairness questions, in particular, Adult and COMPAS are, by far, the most studied datasets by far ([1] Figure 1). Given known issues with the Adult dataset, Folktables (Ding et al.) was developed to provide a wider sample of data for developing policy decisions based on census data. How this much larger sample of census data should be used is a question we seek to answer through our work; as we show, using all available data is not necessarily the best solution.
> We point to Figure 10 in the appendix where we show the benefits of increasing dataset size. We see that accuracy and worst group accuracy both improve while group disparities decrease monotonically with more data in the tabular setting. This motivates our work in questioning whether these benefits of more data can be applied to all states, even the smaller ones like South Dakota.
>
> > In this simpler paradigm with models such as Logistic Regression etc, instead of continuously adding more data for generalizability, would it not make sense to just focus on approaches that curb distribution shift and retrain the model cyclically over certain time periods?
>
> We want to emphasize that state-of-the-art models for tabular datasets are not deep learning models, but tree-based methods and simply MLPs (NN, GB, XBG in Section 5) [2, 3]. In the settings we study, extremely large general tabular datasets are available, but the benefits of incorporating massive tabular data for a specific task in a subset of the data are unknown. Our work follows a line of data-centric works that study the effect of data composition on different state-of-the-art models for the task at hand.
> In fact, our excess KL metric is designed to evaluate distribution shifts induced by new data sources in order to select the best dataset source. Our technique is a version of curbing distribution shift.
>
> **Response to Weakness**
>
> **LLMs and scope of work**:
> > The focus on tabular datasets significantly narrows the paper's scope, especially considering that data scaling is a critical concern in large language models (LLMs) and other domains outside of tabular data (such as NLP and Vision).
>
> Indeed, we agree with the reviewer that running similar experiments on LLMs, and maybe also VLMs would increase the scope of our paper. However, due to resource constraints, pretraining LLMs from scratch with different data sources is currently outside of our capabilities and our scope. However, we emphasize that the folklore of more data being better is ubiquitous in machine learning even outside of the LLM setting. Furthermore, the question of how many EHR, loan approval, and recommendations datasets should be combined is important for practical decision-making. Our paper gives guidance in this common setting in terms of how data composition can be related to scaling matters.
>
> **Generalizability**
> > Furthermore, the results obtained (as well as the approach undertaken) are highly dataset dependent (for e.g., what if I sample the reference test set differently for Folktables? Or use a different state altogether?)
>
> We agree that using a different state altogether might result in different results. However, the contribution of our work is not to claim that “data scaling benefits are **always** eclipsed by distribution shift” (this conclusion cannot be true if $D_{s_2}$ is sufficiently close to $D_{s_1}$). What we are trying to answer is “**How** does data composition impact model performance under scaling” (Figure 1 of main text). Our contribution is to give a principled theoretical model for data accumulation that is flexible to describe effects regardless of the dataset. Our empirical results show a **there exists** claim where we want to urge caution when adding data to illustrate that there are splits of datasets that lead to undesirable effects.

---

> > ### Author Response · Authors · 2023-11-16
> > **Clarification of our contribution and justification of our scope (2/2)**
> >
> > **Evaluation Across All Datasets**:
> > We include the sequential case for all datasets. For other experiments, we focus on the Folktables dataset to ensure our message is conveyed as clearly as possible since there are so many steps (e.g. Mix/Seq KL -> test train KL vs Acc -> excess KL vs acc drop -> data selection vs Acc).  We have additional results for the mixture setting that we can include in the supplementary materials. For Section 5 experiments, if there is a Figure or set of Figures that would motivate the reviewer to improve their rating of our work, we are happy to add them.
> >
> > **Conclusion**
> > We hope our response has clarified and alleviated the reviewer's concerns. If there are further concerns that can be addressed to help improve our work please do not hesitate to inform us.
> >
> >
> > **References**
> > [1] Fabris, A., Messina, S., Silvello, G., & Susto, G. A. (2022). Tackling documentation debt: a survey on algorithmic fairness datasets. In Equity and Access in Algorithms, Mechanisms, and Optimization (pp. 1-13).
> >
> > [2] Kadra, A., Lindauer, M., Hutter, F., & Grabocka, J. (2021). Well-tuned simple nets excel on tabular datasets. Advances in neural information processing systems, 34, 23928-23941.
> >
> > [3] Shwartz-Ziv, R., & Armon, A. (2022). Tabular data: Deep learning is not all you need. Information Fusion, 81, 84-90.

---

> > > ### Comment · Reviewer_4rSz · 2023-11-21
> > > **Response to Rebuttal**
> > >
> > > I would like to express my gratitude to the authors for their rebuttal. However, after going through the response my concerns largely remain. I have condensed some of these below, and would be happy to engage with the authors on them or if I have misunderstood something. If the authors can provide adequate justification, I am amenable to updating my score:
> > >
> > > * **Generalizability**:
> > >     * I am not convinced by the response provided. I understand the research question the authors seek to answer, but when accumulating data at scale, wouldn't practitioners automatically evaluate if the data being added is improving the predictive capability of a model (be measuring performance on a test set)? Also, if one were to employ k-fold cross validation over multiple test set strata, wouldn't this give a holistic picture of whether or not the data being added actually improves the model or if this performance is just specific to the test/validation set at hand? This is my primary concern with generalizability since the research question posed _cannot_ actually be answered in an effective or generalizable way (e.g. if states are different), and an evaluation via cross validation (which is already a part of most data science pipelines) needs to be undertaken irrespective. I also do not think any data science practitioner is just throwing large scale tabular data at these models without evaluating incremental performance gain using the added data.
> > >     * Regarding my previous question in the review, I am still not sure why a practitioner cannot just retrain their model cyclically and curb distribution shift. Basically, keep churning old training data out and adding recent data. If the current test set is sampled from the recent data distribution, this should curb distribution shift. This should ideally work and is also what is likely done in most production pipelines. In my opinion, this should have nothing to do with the models being classical or deep learning based as mentioned in the authors' response.
> > >
> > > * **LLMs and scope of work**: I agree with the authors about the importance of tabular data. But as pointed out, models for tabular data are simpler (and faster) to train, so scaling can simply be evaluated using cross-validation or evaluation on test sets. However, for LLMs (or deep learning models) this is not possible, due to the retraining time complexity. Therefore, where I think the work would be extremely useful is in fine-tuning LLMs or deep learning models (for e.g. for classification on the same datasets considered) and then undertaking the aforementioned scaling analysis. This is what I was suggesting in my review originally as pretraining LLMs or deep models is clearly not a viable option.
> > >
> > > Given the concerns above, I will maintain my current score for now.

---

> > > > ### Author Response · Authors · 2023-11-23
> > > > **Further answers to reviewer concerns (1/2)**
> > > >
> > > > We are encouraged and thank the reviewer for continuing to engage with the discussion of our paper!
> > > >
> > > > **Generalizability**
> > > > > wouldn't practitioners automatically evaluate if the data being added is improving the predictive capability of a model (be measuring performance on a test set)? Also, if one were to employ k-fold cross validation over multiple test set strata, wouldn't this give a holistic picture of whether or not the data being added actually improves the model or if this performance is just specific to the test/validation set at hand?
> > > >
> > > > In a real world scenario, we would expect practitioners to engage in ongoing evaluations of their models. However, in this work, we aim to formalize the nature of that observed impact by adding additional sources, and provide additional intuition for the model performance in different contexts of data accumulation. We hope this can inform practitioner decision-making in contexts and scales where such manual evaluations are less practical.
> > > >
> > > > We will use Folktables as an example. Empirically, we observe diminished accuracy across the reference test set when the model is trained on a different state (e.g., CA, LA, MA) — this is a distribution shift. However, if the model is trained from the same state as the reference test set, and we add to this training set with additional data from other states, it becomes unclear if the advantage of a larger dataset outweighs the distribution shift caused by the addition of this additional source. Thus,  the impact of additional sources of data can yield various, less predictable results depending on the state since the change in the training set does not only involve a distribution shift but also a change in scale (ie.  $n$ has vastly increased). From Figures 9 and 11 in our paper, we see that adding states (with lower excess KL Equation 1) results in better scaling results. Evaluating training sets composed of single sources is insufficient to validate what happens when we grow our original data by adding the same data source; particularly for questions like robustness and disparity.
> > > >
> > > > >This is my primary concern with generalizability since the research question posed cannot actually be answered in an effective or generalizable way (e.g. if states are different), and an evaluation via cross-validation (which is already a part of most data science pipelines) needs to be undertaken irrespective.
> > > >
> > > > In an age of increasingly heterogeneous and large-scale datasets, there is an urgent need for frameworks that characterize data composition.  In contrast to prior model-centered approaches, we create a framework to characterize data composition as it changes due to scaling, a phenomenon we name data accumulation. In this version of the study, we focus on characterizing this phenomenon in the simple setting of tabular data, with the hope to include future work where we can discuss the consequences of such data-driven decision-making in more complex settings.
> > > >
> > > > Regarding the comment of our investigations needing to include a wider range of data sources (eg. greater diversity of states), we understand that the scope of our current experiments is limited to tabular settings, we believe our experiments are still sufficient to demonstrate the need for further study on data accumulation within the machine learning field at large. We feel that the tabular setting provides an illustration of phenomena that will hopefully spur further investigations into other settings.
> > > >
> > > > > I also do not think any data science practitioner is just throwing large-scale tabular data at these models without evaluating incremental performance gain using the added data.
> > > >
> > > > Indeed, we want the primary impact of our work to be a cautionary tale against blindly throwing large-scale tabular datasets into models without careful analysis of the effects of the data composition. However, this does indeed seem to be the default practice. Many practitioners include entire tabular datasets in their models with the underlying assumption that increasing the size of their training data, often without examining the heterogeneity of underlying data sources,  will improve the performance of models. As one example, the vast majority of benchmarks treat datasets as a monolithic source without more granular analysis, e.g., MIMIC-III benchmarks in [1], Yelp RecSys benchmarks [2] .

---

> > > > > ### Author Response · Authors · 2023-11-23
> > > > > **Further Answers to Reviewer Concerns (2/2)**
> > > > >
> > > > > > Basically, keep churning old training data out and adding recent data. If the current test set is sampled from the recent data distribution, this should curb distribution shift.
> > > > >
> > > > > If the test set is the most current data, the reviewer's suggestion could be helpful. However, there are two main limitations. (a) This approach may increase distribution shift if there are abrupt changes. For example, in Yelp, if we cyclically included new data from the same state during the pandemic, when restaurants open finally for in-store service, older data is actually more relevant than more recent data. (b) Many surveys and data streams do not have temporally rich collections. For example, the Census is conducted every 10 years, and MIMIC-III and MIMIC-IV have a 7-year gap in release date and a huge mismatch in available features (ICD codes also changed). Thus, we can imagine it may be more useful to have a direct characterization of distributional differences to be used for decision-making agnostic (but encompassing) temporal shifts; focusing on tabular data gives us this advantage.
> > > > >
> > > > > Also, there are various application contexts where the need for performance improvement is more immediate, or the opportunity to collect data at a later date is not there. In such cases, practitioners tend to be more likely to seek alternative additional data sources to supplement the existing training set.
> > > > >
> > > > > **LLM and scope of work**
> > > > > > Therefore, where I think the work would be extremely useful is in fine-tuning LLMs or deep learning models (for e.g. for classification on the same datasets considered) and then undertaking the aforementioned scaling analysis.
> > > > >
> > > > > Several prior works including [3] show that the majority of information retained occurs in the pre-training stage; this led us to believe training from scratch would be necessary for a convincing demonstration. The choice to focus these sets of experiments on the tabular setting was intended to illustrate the data accumulation phenomenon within **one** practical setting in a modality prevalent for large datasets. The studied context is particularly relevant to to practitioners who consider data composition questions in primarily tabular formats, e.g., structured healthcare data, recommender systems, climate weather data, and government surveys. We agree that other data modalities and corresponding models (e.g., LLMs) will be exciting to study in future work, and we look forward to researchers building on our data composition model.
> > > > >
> > > > > In summary, we hope our responses have alleviated the reviewer's concerns and hopefully shine a light onto a much-needed data-centric perspective that we are providing. Thanks again for the engagement!
> > > > >
> > > > > **References**
> > > > >
> > > > > [1] Harutyunyan, H., Khachatrian, H., Kale, D. C., Ver Steeg, G., & Galstyan, A. (2019). Multitask learning and benchmarking with clinical time series data. Scientific data, 6(1), 96.
> > > > >
> > > > > [2] Zhu, J., Dai, Q., Su, L., Ma, R., Liu, J., Cai, G., ... & Zhang, R. (2022, July). Bars: Towards open benchmarking for recommender systems. In Proceedings of the 45th International ACM SIGIR Conference on Research and Development in Information Retrieval (pp. 2912-2923).
> > > > >
> > > > > [2] Zhou, C., Liu, P., Xu, P., Iyer, S., Sun, J., Mao, Y., ... & Levy, O. (2023). Lima: Less is more for alignment. arXiv preprint arXiv:2305.11206.

---

### Official Review · Reviewer_UNBZ · 2023-11-01

**Soundness:** 3 good
**Presentation:** 3 good
**Contribution:** 3 good
**Rating:** 6
**Confidence:** 4

**Summary:**

Describes how the way data is added to a model (data accumulation) affects performance against the reference test set. Two methods are presented. First, a data mixture method where data is added in the same subgroup mixture as the original dataset. Second is sequential - this is where datasets are added one after another with no guarantee that the mixture of subgroups is the same as the original data. The authors point out that sequential additions can harm model performance especially when there are distinct distribution differences between the datasets (i.e. high KL divergence).

**Strengths:**

Well written and useful analysis, and especially suitable for this track.  It guides researchers on what to expect when adding new datasets, what circumstances lead to good outcomes and which one might not. Also great caution to the assumption that more data (except perhaps noisy or corrupt data etc) is always good for the model.

**Weaknesses:**

Overall this was an interesting paper to read. Most of these are about clarifications and how the authors have interpreted their results.

It is unclear how the target dataset is constructed. It should not matter in the mixture set-up but it would be consequential in the sequential set-up. The target set should be a sample from all n datasets, unless it is updated each time a new dataset is added.
It is also not clear how long the model is re-trained with the new examples. This can help us better understand if the examples can’t be learned or if the model just did not have as many iterations to incorporate these new examples.
The implications of this work are not clear. In real-world settings, if there exists a datasets similar to one that we currently have but has high divergence does it mean it should not be included in the analysis? Doesn’t not doing so restrict the model from better generalising? Eg. Yelp reviews in MN vs SD.
Thirdly, it looks like adding more data reduces performance disparity between groups and in general helps the least performing group. Reducing disparity is perhaps indicating that the model is generalising better and getting more robust and these should be good things.

**Questions:**

1. How is the reference test set constructed? If it's in the appendix, it should included in the main paper because it is consequential.
2. How long do you retrain after adding new datasets?

---

> ### Author Response · Authors · 2023-11-18
> **Answers to reviewer questions and clarifications**
>
> We thank the reviewer for taking the time to read and understand our work. We agree with the reviewer that our work is useful in the datasets track – to help formalize practical questions raised by the distribution shift that arises from data scaling.
>
> **Answering Reviewer Questions**
> 1. The reference test set is constructed in our experiments from a very small initial training source $D_{S_1}$ (e.g. South Dakota). We make this assumption to simulate a setting where there is some limited access to the initial dataset of interest but it is then unclear if adding more data would help with outcomes on this initial dataset. We assume that the eventual deployment set might be different from the first source but is likely most similar to the first source.
> 2. We train from scratch at each dataset size presented in our plots. Thus, each training trajectory shows 5 models over 50 seeds (not 5 - typo in the main text) at for each size n.
>
> **Implications of our work**
> > The implications of this work are not clear. In real-world settings, if there exists a datasets similar to one that we currently have but has high divergence does it mean it should not be included in the analysis? Doesn’t not doing so restrict the model from better generalising? Eg. Yelp reviews in MN vs SD.
>
> In the real world, if we knew a dataset is vastly different (e.g. loan applications for credit cards vs loan applications for mortgages), it does make sense to not include it. However, the question that arises is, what level of divergence would harm my model, and what level is acceptable? In all of our datasets, we are looking at the differences that might arise **within** a dataset. We show that adding data from states (near and far) with reasonable divergences (e.g. there is SOME shift) improves outcomes. Especially when the starting dataset is small (and undiverse) excluding data in the same dataset might not be the best choice. This is a very nuanced point and we will better explain it in our paper.
>
> > Thirdly, it looks like adding more data reduces performance disparity between groups and in general helps the least performing group. Reducing disparity is perhaps indicating that the model is generalising better and getting more robust and these should be good things.
>
> Reducing performance disparity by adding more data is indeed a desirable goal. We hope to show that by measuring not just overall accuracy but also disparity and worst group performance, understanding the consequences of data scaling is a multi-faceted pursuit and KL divergence as a lone metric is only just a first heuristic. We hope our work will inspire future work to develop further metrics for determining when to add data for properties like disparity and worst group robustness in particular.
>
> We hope our responses have clarified the questions raised. If we can further provide any explanations to improve the reviewers confidence in our work, please do not hesitate to raise further questions.

---

### Official Review · Reviewer_r5EZ · 2023-11-01

**Soundness:** 2 fair
**Presentation:** 3 good
**Contribution:** 2 fair
**Rating:** 6
**Confidence:** 4

**Summary:**

The authors model data accumulation from multiple sources and present an analysis of two strategies that result in adding more data, degrading the overall model performance. They empirically demonstrate on three real-world datasets that adding training data can reduce overall accuracy and reduced worst-subgroup performance while introducing further accuracy disparities between subgroups.

**Strengths:**

- the authors tackle the well-known issue that more data does not always lead to better machine learning outcomes: data quality (the whole dataset composition should mirror the data we will receive at inference time) is of primary importance.
 - the paper is of good quality: the authors propose several scenarios and work with real-world data to draw conclusions
 - the paper is well-structured and written

**Weaknesses:**

- the authors did not consider research on domain adaptation, which could be considered key in this particular setting
 - the authors did not check for data-based techniques used in active learning settings that can help identify data relevant to machine learning models

**Questions:**

We consider this research interesting and relevant. Nevertheless, we would like to point to the following improvement opportunities:
1. "training data is often considered to be set in stone and imposed as a pre-existing and static constraint." -> The authors should consider that while sometimes this is true, the fact that distribution shift exists and takes place should be, therefore, evaluated on training sets too. We encourage you to reframe the sentence stating such an evaluation as a best (and often forgotten) practice.
2. *Criteria for rejecting more data*: The problem posed by the authors resembles active learning and some specific data-based strategies. Furthermore, some research has been performed on active learning and stopping criteria. The authors may be interested in researching these areas. Here, we list two works they may find useful: (a) Fu, Yifan, Xingquan Zhu, and Bin Li. "A survey on instance selection for active learning." Knowledge and information systems 35 (2013): 249-283, and (b) Zhang, Yexun, et al. "Stopping criterion for active learning with model stability." ACM Transactions on Intelligent Systems and Technology (TIST) 9.2 (2017): 1-26.
3. *Experimental setup*: we consider the experiments valuable and valid. Nevertheless, the authors should consider enriching them with some scenarios where domain adaptation is used to mitigate distribution differences. The authors may be interested in the following work: Farahani, Abolfazl, et al. "A brief review of domain adaptation." Advances in data science and information engineering: proceedings from ICDATA 2020 and IKE 2020 (2021): 877-894.
4. *Results and analysis*: Do the authors venture some hypothesis as to why the XGB model is robust to data from different distributions, suffering a lower accuracy loss?

---

> ### Author Response · Authors · 2023-11-18
> **Responses to reviewer questions**
>
> We thank the reviewer for their helpful suggestions on our work and we value their positive comments on the quality and presentation of our work.
>
> Please see our answers to reviewer questions below:
> 1. This is a good point, we will indeed rephrase this sentence to better acknowledge prior work in the distribution shift (in addition to our discussion in C.5 of our supplementary materials).
>
> 2. We thank the reviewer for pointing out related work in active learning. We agree that it is important and we will include this in our work. We do highlight that our KL metric is distributional rather than for specific instances as described in equation 15 of Fu et. al. While Zhang et. al. takes a, notably different, model change approach to stop data addition, we believe it is still very wonderful prior work we will include to frame our contribution.
>
> 3. In our experimental setup, we specifically wanted to focus on data interventions rather than model interventions suggested by domain adaptation strategies. In our supplementary materials C.5 we discuss in detail our work in relation to domain adaptation in particular.
>
> 4. Excellent question. While in our results we show that XGBoost does not suffer from an **accuracy drop**, the model still performs much worse than it would if more data from a more similar distribution were to be added. In Figures 11b and 11c (supplementary materials), we show that XGB continues to improve substantially when other sources are added like Louisiana.
>
> We value these helpful comments for the reviewer. We believe that taking this data composition framework to understand scaling under possible distribution shifts is an important direction that will inspire future work in richer domains. We are happy to answer any further questions or add experiments and hope that the reviewer will vouch for our work!

---

> > ### Comment · Reviewer_r5EZ · 2023-11-23
> >
> > We have reviewed the authors' responses and have no further observations.

---

### Author Response · Authors · 2023-11-18
**Response to All Reviewers**

We thank the reviewers for their thoughtful feedback. In particular, we want to highlight that reviewers found our paper to be of good quality [r53Z], well written with useful analysis [UNBZ], work(ed) with real-world data to draw conclusions [r53Z], and overall an interesting paper to read [UNBZ].

Here we highlight key themes that appear across the reviewers.

**The high-level question: How does training data composition impact model performance under scaling?**
- We emphasize that our work focuses on tradeoffs between dataset size and potentially lowering data quality. We give motivation for this common scenario in Sections 1-2 of our work. Under our data composition framework, it is not sufficient to simply observe that different sources (e.g., states, ICU admission) may have a different distribution from a reference test set of interest. We seek to formalize the mechanisms that contribute to data composition changes as the number of data points increases.
- Our work presents a data-centric perspective on the machine learning process. Our novel contribution is formalizing models to describe choices around data rather than choices around the algorithm. The empirical experiments we present in Section 5, are a proof of concept for the theory we propose in Section 3; showing that there exist sources (within the same dataset) that would be better to not include. We believe this is an incredibly important angle in an era where more data is seen as the blanket solution to improve any type of machine learning model.

**Justification for the current scope of experiments:**
> Reviewer 4rSz questioned whether tabular data is an important setting.
- We reiterate that many practical decision-making tasks which use machine learning are based on tabular data. For example, loan decisions (e.g., German Credit), government resource allocation (e.g., ACS/Census), recommender systems (e.g., Yelp), recidivism prediction (e.g., COMPAS), and precision health care (e.g., MIMIC-IV) decisions all rely on tabular datasets. Particularly for high-stakes tasks with major fairness and equity concerns, tabular datasets have been the standard benchmarks [1].
- Our proposed models of data accumulation (e.g., Section 3) can be easily applied to other domains in future work. In particular, KL-based heuristics (without a scale dimension) have been applied for data selection in LLMs with a different density estimation procedure (Xie et. al. in our main paper). However, we believe better characterization of datasets before they are combined can benefit a variety of different tasks beyond the scope of our paper.

**Evaluation setup and how it differs from distribution shift**
> Reviewers 7keK and 4rSz both had some confusion about how our problem differs from distribution shift.
- [**data vs algorithmic interventions**] The vast majority of distribution shift literature aims at better test set performance given some fixed shift between the test and train dataset (e.g. WILDs benchmarks). In practical scenarios, the distribution shift is not fixed but a byproduct of data choices. There are likely multiple sources that can be drawn from and the decision must be made in terms of which sources to include - this then results in changes to the training dataset that then impacts model performance. Furthermore, it is likely that as more and more data sources are included, there might be variance in the quality of these sources and there are no existing measures to evaluate true data quality. Thus, this data selection problem is not trivial. For simplicity, our test distribution is one of the sources, but the same model applies when the test distribution is not one of the potential training data sources.
- [**distribution shift vs n**] The setup of the canonical distribution shift task implies a fixed dataset size since the proposed algorithms in prior works are model-based interventions. We model how dataset composition may introduce distribution shifts and thus compromise scaling properties. To the best of our knowledge, $n$ as a causal mechanism for distribution shift has not been studied.
- [**beyond accuracy drop**] The standard metric for the efficacy of algorithms for distribution revolves around accuracy drop. Our goal is entirely different in that we are evaluating the claims put forth by prior work that more data always improves accuracy, robustness, and fairness. Our results show a resounding no, especially in comparison to adding better sources of data (Figure 11). Moreover, we provide a notion of “better” data that is intricately tied to our model of data accumulation (Section 3).

---

> ### Author Response · Authors · 2023-11-18
> **Summary**
>
> In summary, we rigorously study the question of when and why data composition matters when accumulating training data for improved model performance. Our question is practically grounded in many real-life scenarios where the cleanest source of data is very small, and care must be taken on when and how to supplement the available training dataset for improved performance. We hope this clears up concerns raised by all the reviewers, and we thank the reviewers for engaging this scholarly process with us by carefully reading our paper and considering our contributions.
>
> **References**
> [1] Fabris, A., Messina, S., Silvello, G., & Susto, G. A. (2022). Tackling documentation debt: a survey on algorithmic fairness datasets. In Equity and Access in Algorithms, Mechanisms, and Optimization (pp. 1-13).

---

### Meta-Review · Area_Chair_XofP · 2023-12-03

**Metareview:**

The paper studies data compositions from multiple sources and how adding more data can help or hurt performance.
Reviewer r5Ez (6) find the research interesting and relevant, nothing particular to say.

* Reviewer UNBZ (6) finds the paper to provide useful analysis, in particular for the data track, in particular the warning that adding data doesn't always improve performance. However, the reviewer noted that is is unclear how the reference dataset is construction, and some implications of the work are not clear. The reviewers clarified the construction and implications.

* Reviewer 4rSz (3) appreciates the theoretical results, and evaluation on three real world datasets. However, the reviewer finds multiple issues with the work: a lack of generalizability, inconsistent evaluation across datasets, and a limited scope and applicability (i.e., limited to tabular data). After the discussion; I agree with the authors that studying tabular data alone is sufficient and LLMs are out of scope, however I agree with the reviewer on the generalizability issues.

* Reviewer 7keK (3) finds the results to be known in the research community, sees a lack of novelty, and finds the problem statement unclear. The authors point out novelties including how distribution shifts can arise when scaling data. I agree with the authors that the paper includes new results and that the problem studies is clear.

While the paper certainly has value and has promising results, I also think that the results are relatively specific to a narrow setup studied, and that more work on realistic setups is required for acceptance.

**Justification For Why Not Higher Score:**

The results are relatively specific to a narrow setup studied, and that more work on realistic setups is required for acceptance.

**Justification For Why Not Lower Score:**

N/A

---

### Decision · Program_Chairs · 2024-01-16

Reject